# Artificial intelligence exceeds humans in epidemiological job coding

Mathijs A. Langezaal [1,2✉], Egon L. van den Broek [2✉], Susan Peters[3], Marcel Goldberg [1], Grégoire Rey [4], Melissa C. Friesen[5], Sarah J. Locke[5], Nathaniel Rothman[5], Qing Lan [5] & Roel C. H. Vermeulen [3]

## Abstract

**Background** Work circumstances can substantially negatively impact health. To explore this, large occupational cohorts of free-text job descriptions are manually coded and linked to exposure. Although several automatic coding tools have been developed, accurate exposure assessment is only feasible with human intervention.

**Methods** We developed OPERAS, a customizable decision support system for epidemiological job coding. Using 812,522 entries, we developed and tested classification models for the Professions et Catégories Socioprofessionnelles (PCS)2003, Nomenclature d'Activités Française (NAF)2008, International Standard Classifications of Occupation (ISCO)-88, and ISCO-68. Each code comes with an estimated correctness measure to identify instances potentially requiring expert review. Here, OPERAS' decision support enables an increase in efficiency and accuracy of the coding process through code suggestions. Using the Formaldehyde, Silica, ALOHA, and DOM job-exposure matrices, we assessed the classification models' exposure assessment accuracy.

**Results** We show that, using expert-coded job descriptions as gold standard, OPERAS realized a 0.66–0.84, 0.62–0.81, 0.60–0.79, and 0.57–0.78 inter-coder reliability (in Cohen's Kappa) on the first, second, third, and fourth coding levels, respectively. These exceed the respective inter-coder reliability of expert coders ranging 0.59–0.76, 0.56–0.71, 0.46–0.63, 0.40–0.56 on the same levels, enabling a 75.0–98.4% exposure assessment accuracy and an estimated 19.7–55.7% minimum workload reduction.

**Conclusions** OPERAS secures a high degree of accuracy in occupational classification and exposure assessment of free-text job descriptions, substantially reducing workload. As such, OPERAS significantly outperforms both expert coders and other current coding tools. This enables large-scale, efficient, and effective exposure assessment securing healthy work conditions.

## Plain language summary

Work can expose us to health risks, such as asbestos and constant noise. To study these risks, job descriptions are collected and classified by experts to standard codes. This is time-consuming, expensive, and requires expert knowledge. To improve this coding, we created computer code based on Artificial Intelligence that can both automate this process and suggest codes to experts, who can then check and change it manually if needed. Our system outperforms both expert coders and other available tools. This system could make studying occupational health risks more efficient and accurate, resulting in safer work environments.

[1] Population-Based Epidemiological Cohorts Unit UMS11, INSERM, 16 Avenue Paul Vaillant Couturier, Paris 94807 Villejuif, France. [2] Department of Information and Computing Sciences, Utrecht University, Princetonplein 5, Utrecht, 3584CC Utrecht, The Netherlands. [3] Institute for Risk Assessment Sciences, Utrecht University, Yalelaan 1, Utrecht, 3584CL Utrecht, The Netherlands. [4] Center for Epidemiology on Medical Causes of Death (CépiDc), INSERM, Le Kremlin-Bicêtre, France. [5] Occupational and Environmental Epidemiology Branch, Division of Cancer Epidemiology and Genetics, National Cancer Institute, Bethesda, MD, USA. ✉email: m.a.langezaal@uu.nl; vandenbroek@acm.org

Occupation is a major component of adult life and is an important determinant of overall health. Workers are exposed to negative conditions such as prolonged sitting, stress, and chemical and physical agents including diesel engine exhaust and asbestos[1]. Such occupational exposures globally account for 2.8% of deaths and 3.2% of disability-adjusted life years from all causes[2]. For example, about 26% of low back pain has been estimated to be work-related and 3.2% to 4.6% of all cancer deaths are due to occupational exposure. Occupational epidemiological studies that assess exposure-response associations fuel interventions to promote workers' health. Such studies rely on high-quality exposure assessments to secure correct or unbiased results[3].

Occupational data in population-based cohorts is collected from respondents via open-ended surveys where they report their job title as they know it in a free-text format[4]. Subsequently, Job-Exposure Matrices (JEMs) can be used to assign a wide range of exposures to occupations[5]. A JEM comprises of estimated probabilities and intensities of exposure to harmful agents by standardized occupational titles and industries. Through the use of occupational and industry classification systems, free-text job descriptions are translated into such standardized occupational titles and industries[6].

Occupational coding is performed manually by expert coders. To ensure high quality and consistency of coding, documentation, procedures, and training are provided[4]. Nevertheless, consistent coding of job descriptions remains challenging, given the large number of different outcome categories. After three months of extensive coding and training, the coding efficiency of an expert coder can reach ~2700 codes per month[7]. Furthermore, because of the repetitive nature of the task, prolonged occupational coding sessions result in reduced coding accuracy due to fatigue. Hence, given the large scale of many occupational epidemiological cohorts, multiple expert coders need to be deployed. This introduces a 44–89% inter-coder reliability range for the most detailed coding level[8], with more recent studies reporting ranges between 42–71%[9–12].

Tools have been developed to support manual assignment of occupational codes to job descriptions[13,14]. Additionally, tools have been developed that enable completely automatic coding, achieving a prediction accuracy ranging 15–64% on the highest coding level with an out-of-distribution accuracy ranging 17–26% (see Table 1)[10,15–20]. Although some coding tools achieve a human-coder level prediction accuracy, their exposure assessment accuracy may be lower. Research on automatic coding tools reports inter-coder reliability of 51% with an exposure assessment Cohen's Kappa score between 0.4–0.8[7]. However, a separate study reports higher minimum human exposure assessment Cohen's Kappa ranges between 0.66 and 0.84 with only 36–50% inter-coder reliability[21]. Hence, human intervention is required during or after the automatic coding process to ensure reliable exposure assessment. However, current (semi-)automatic coding tools either lack the implementation of machine learning techniques, which are essential for achieving human-level prediction accuracy[16,22] or are limited to one national occupational coding classification system[10]. This implies that only JEMs using these national classifications can be applied, limiting the use of these tools in global exposure assessment studies requiring other classifications. Consequently, we developed, tested, and validated OPERAS, a decision support system for epidemiological job coding that utilizes ML-based classification models for four (inter)national classification systems. Its goal is to support expert coders during occupational coding through both automatic coding as well as decision support through code suggestions. Our evaluation shows that OPERAS outperforms existing automatic coding tools and expert coders, leading to a substantial workload reduction and highly accurate exposure assessment.

## Methods

OPERAS has been developed via six phases. First, we acquired datasets and identified classification performance barriers. Additionally, we specify the origin, available input classes, and used occupational classification systems. Second, to enable complete code suggestions, we removed missing and incomplete codes during data preparation. Furthermore, to ensure optimal classification performance, we reduced the dimensionality and retained crucial information of the free-text job descriptions through multiple Natural Languange Processing (NLP) techniques. Third, using state-of-the-art gradient tree boosting, we trained the classification models. We optimized for generalizability by using pre-defined parameter settings and the inclusion of generally described job descriptions from the coding index in the training set. Fourth, using prediction accuracy and Cohen's kappa, we evaluated the classification models on their classification performance and present its impact on the occupational coding process. Fifth, based on two occupational inter-coder reliability studies, we compared OPERAS' inter-coder reliability to expert coders. Sixth, using four JEMs, we evaluated the NAF, PCS, ISCO-88, and ISCO-68 classification models on their exposure assessment for two groups: 1) all individuals, and 2) exposed individuals. In the following subsections, each phase will be explained in detail.

**Datasets**. Using French[23], Asian[24], and Dutch[25] datasets, we developed OPERAS' classification models. These datasets respectively contain French, English, and Dutch manually expert-coded free-text job descriptions in four hierarchically structured occupational/activity sector coding classifications. Each classification consists of multiple numbers and/or letters (e.g., "211A") where each added (combination of) character(s) provides additional detail about the job description. For example, the major, sub-major, minor, and unit groups of the ISCO-88 code "2221" are "Professionals" (2), "Life Sciences and Health Professionals" (22), "Health Professionals" (222), and "Medical Doctors" (2221), respectively. For insufficiently detailed job descriptions the expert coder will replace not codable levels with a pre-defined character (e.g., "#"). In the following paragraphs, each dataset and coding classification will be further described. The descriptive statistics of each dataset can be found in Tables 2 and 3.

Constances is a general-purpose cohort of French adults aged 18–69 with a focus on occupational and environmental factors[23]. The dataset contains free-text answers to open-ended questions regarding participants' occupations held and sector of activity. This information was manually coded into the French Nomenclature d'Activités Française (NAF)2008 and Professions et Catégories Socioprofessionnelles (PCS)2003 classifications by expert coders using a dedicated application showing the entire summary of a participant's occupational history[26]. Here, a NAF code describes one's activity sector and consists of four digits and one letter (e.g., "4231A"), whereas a PCS code describes occupation and consists of three digits and a letter (e.g., "211A"). The Constances cohort has been reviewed and approved by the Institutional Review Boards of the French Institute of Medical Research and Health, the "Commission Nationale de l'Informatique et des Libertés", and the "Conseil National de l'Information Statistique". Written informed consent was obtained from all participants. Access and permission to reuse the restricted data for the current study were provided by the Principal Investigators (PIs) of the original study.

The AsiaLymph dataset is a hospital-based case-control study of lymphoma among Chinese in Eastern Asia[24]. This dataset contains free-text traditional and simplified Chinese occupations, tasks, employers, and products that were translated into English. They were manually coded using the International Standard

**Table 1 Supported occupational classification(s), applied classification algorithm, and accuracy (%) of the currently available automatic occupational coding tools.**

| Coding tool | Occupational Classification(s) | Classification algorithm | Accuracy | |
|---|---|---|---|---|
| | | | CL (OC) | Acc. |
| ACA-NOC[15] | Canadian NOC2016 | Multiple search strategies | 1 (10) | 82 |
| | | | 2 (40) | 72 |
| | | | 3 (140) | 65 |
| | | | 4 (500) | 59 |
| CASCOT[16] | UK SOC(2010, 2000, 90), UK SIC(2007, 2003, 92, 80), ISCO-08 | String similarity | 4 (412) | 64[a] |
| NIOCCS[17] | US SOC2010 | Deterministic supervised machine learning algorithm | 1-2 (23) | 45 |
| | | | 3 (97) | 22 |
| | | | 4-5 (461) | 19 |
| | | | 6 (840) | 15 |
| Procode[18] | French PCS2003, French NAF2008 | Complement Naive Bayes | PCS2003: | |
| | | | 1 (8) | 81 |
| | | | 2 (24) | 73 |
| | | | 3 (42) | 70 |
| | | | 4 (497) | 57 |
| | | | NAF2008: | |
| | | | 1 (21) | 82 |
| | | | 2 (88) | 79 |
| | | | 3 (272) | 68 |
| | | | 4 (615) | 66 |
| | | | 5 (732) | 63 |
| SOCcer[10] | US SOC2010 | Stacked ensemble classifier | 3 (97) | 64 |
| | | | 6 (840) | 51 |
| SOCEye[7] | US SOC2010 | Global and local classification approaches | 1-3 (97) | 85 |
| | | | 4-5 (461) | 72 |
| | | | 6 (840) | 51 |

Accuracy is given for each coding level (CL) and occupational classification specified in the original study. The number of outcome categories (OC) is specified for each coding level.
[a]Reports on CASCOT claim that 80% of the suggested codes have a confidence score of 40 or higher, of which 80% are correct. However, the used occupational classification, methods, and detailed results are not mentioned and can thus not be verified.

**Table 2 Descriptive statistics of viable entries from the Constances, AsiaLymph and Lifework datasets (IQR = interquartile range).**

| Statistic | Constances | | AsiaLymph | Lifework |
|---|---|---|---|---|
| | NAF | PCS | ISCO-88 | ISCO-68 |
| Original nr. of entries | 637,148 | 637,148 | 36,179 | 12,120 |
| Nr. of viable entries | 281,418 | 483,090 | 36,007 | 12,007 |
| Nr. of viable outcome categories | 732 | 497 | 389 | 653 |
| Nr. of input classes | 2 | 2 | 4 | 3 |
| Min. class size of outcome category | 1 | 1 | 1 | 1 |
| Max. class size of outcome category | 77,639 | 27,103 | 2251 | 1220 |
| Median. class size of outcome category (IQR) | 64    (196) | 471    (1003) | 26    (69) | 3    (8) |

**Table 3 Minimum, 25%, 50%, 75% quantiles, and maximum word count of job descriptions per input class for AsiaLymph, Constances, and Lifework datasets.**

| Word count | Constances | | AsiaLymph | | | | Lifework | | |
|---|---|---|---|---|---|---|---|---|---|
| | Profession | Sector | Occupation | Task | Employer | Product | Job name | Job description | Company |
| Minimum | 1.0 | 1.0 | 1.0 | 1.0 | 1.0 | 1.0 | 1.0 | 1.0 | 1.0 |
| 25% quantile | 1.0 | 1.0 | 1.0 | 2.0 | 2.0 | 2.0 | 1.0 | 2.0 | 1.0 |
| 50% quantile | 2.0 | 1.0 | 2.0 | 2.0 | 3.0 | 2.0 | 1.0 | 3.0 | 1.0 |
| 75% quantile | 3.0 | 2.0 | 2.0 | 4.0 | 4.0 | 3.0 | 2.0 | 5.0 | 2.0 |
| Maximum | 14.0 | 16.0 | 24.0 | 21.0 | 19.0 | 27.0 | 15.0 | 24.0 | 13.0 |

Classifications of Occupation (ISCO)-88 coding classification, which consists of four hierarchical levels of detail, each represented with a number (e.g., "4521"). The occupational data were independently coded by two study centers, where discordant codes were resolved by a third expert coder. The AsiaLymph study was approved by the institutional review boards at each of the four participating sites, the U.S. National Institutes of Health, and the U.S. National Cancer Institute. Written informed consent was obtained from all participants. Access and permission to reuse the restricted data in the current study were provided by the PIs of the original study.

The Lifework cohort is a large federated prospective cohort from the Netherlands that quantifies the health effects of occupational and environmental exposure[25]. The dataset includes job names, descriptions, and company types that were manually coded into ISCO-68 occupational codes, which is an earlier version of ISCO-88. This version has an additional level of detail to describe a job description (e.g., "1-21.10"). Contrastingly, instead of supplemented incomplete codes, only the levels that can be coded are given (e.g., "1-21" instead of "1-21.##"). The coding was performed by expert coders, where uncertain initial codes were resolved by a second expert coder (see manual coding procedures in the Supplementary Methods). The Lifework cohort and contributing subcohorts were reviewed and approved by the committee at the University Medical Center Utrecht, the committee at TNO (Dutch Organization for Applied Scientific Research) Nutrition and Food Research, and the committee at the Netherlands Cancer Institute. Participants signed an informed consent form for each subcohort prior to enrolment. Access and permission to reuse the restricted data in the current study were provided by the steering group of the Lifework study.

**Data preparation**. The Constances, Asialymph, and Lifework datasets contain 637,148, 36,179, and 12,120 entries, respectively. To ensure complete code suggestions, entries with incomplete codes were considered nonviable for classification model training. After data cleaning, 281,418, 483,090, 36,007, and 12,007 viable entries remained to train the NAF, PCS, ISCO-88, and ISCO-68 models, respectively. A large number of incomplete NAF and PCS entries were removed using this strategy due to the insufficiently detailed descriptions before occupational coding. However, this was warranted as it ensures optimal decision support during the occupational coding process through complete code suggestions.

As all data were collected in an open-ended format, tokenizing, removing punctuation marks, converting all input to lowercase, and stemming all words to their root form allowed for the reduction of the random noise and dimensionality of the data[27]. Compared to other automatic text-classification systems, we noticed a low word count of the descriptions, and high variability between descriptions of the same occupation[27–29]. Hence, to allow for the retention of important information and similar representation of similar job descriptions, we embedded all descriptions into numerical feature vectors using sentence embedding[30]. Sentence embeddings are fixed-length vector representations of sentences that allow for comparison and computation with other sentences. These embedders have been trained on large corpora of textual data to learn highly generic semantic relations between sentences. The goal is to encode the sentence so that similar sentences (in terms of meaning) are close in the embedding space, and dissimilar sentences are far apart. In the current context, this would result in similar job descriptions being represented as such. When multiple input classes per occupational code were required for classification model training (e.g., job description and sector), we summed the resulting feature vectors to combine all information[31]. To increase the

generalizability of the models, we randomly split up the prepared data from each dataset into a training (60%), test (30%), and validation (10%) set, and supplemented the training set of each occupational classification with job descriptions from their respective coding indexes.

As classification models tend to overly represent majority classes, the class sizes of the outcome categories should be balanced[32]. However, algorithms that synthetically provide data balance could not be applied to the current datasets due to the difference in class size being too large[33,34]. Oversampling algorithms could not compute synthetic samples due to the minority classes having too few samples. Contrastingly, under-sampling resulted in the loss of too much information due to the majority classes being reduced to only a few samples.

**Classification model training**. Using the state-of-the-art gradient tree boosting algorithm XGBoost, we developed the NAF, PCS, ISCO-88, and ISCO-68 classification models[35] (see Fig. 1). We chose this algorithm due to a study performed by Schierholz and Schonlau[22], where multiple Machine Learning (ML) algorithms were evaluated on their classification performance based on five occupational datasets. They found that, when the training data was supplemented with the job descriptions from the coding index, the XGBoost algorithm performed best.

XGBoost generates an ensemble of weak Classification and Regression Trees (CARTs) and combines their predictions to produce a strong, accurate model[35]. The model is trained by minimizing a regularized objective function, which includes a loss function to measure the error between the target class (i.e., the 'gold-standard' occupational code), and a regularization term to prevent overfitting. For multi-class problems such as the current one, XGBoost builds a binary outcome ensemble for each class. The ensemble is trained in an additive manner, where a new CART is added to the ensemble in each iteration to fit the residual errors of the previous iteration. Furthermore, XGBoost uses several techniques to prevent overfitting, such as the addition of the regularization term, subsampling of training data and columns, and the penalization of large weights. These strategies can be fine-tuned using several adjustable hyperparameters. A more extensive description of the algorithm and its hyperparameters can be found in the Supplementary Methods and in the original paper from Chen and Guestrin[35].

Schierholz and Schonlau[22] empirically optimized the hyperparameters for the largest occupational dataset and proposed a default setting. As the datasets used in ref. [22] are similar to the current datasets in terms of data collection (i.e., free-text job descriptions), (number of) outcome categories (i.e., occupational codes), and average word count, we expected similar relative performance when the proposed default set of hyperparameters is used in the current study[36]. Following the instructions from Schierholz and Schonlau[22] we have set lower values for $\eta$ and *max_depth* for optimal classification performance. Hence, we adopted and used the following hyperparameter settings for the development of each classification model: Early stopping if performance does not improve for 1 round, $\eta = 0.6$, *max_delta_step* $= 1$, *max_depth* $= 20$, $\gamma = 1.5$, $\lambda = 1e^{-4}$, *min_child_weight* $= 0$, *subsample* $= 0.75$, *colsample_by_tree* $= 1$, and *colsample_by_level* $= 1$.

We trained the NAF, PCS, and ISCO-68 classification models in the original language of the corresponding dataset, whereas we trained the ISCO-88 model using its English translation. Here, the classification models were solely trained using the training data. The validation set was used during training to allow for early stopping of training to reduce the chance of overfitting[37]. We performed feature selection to find the combination of input

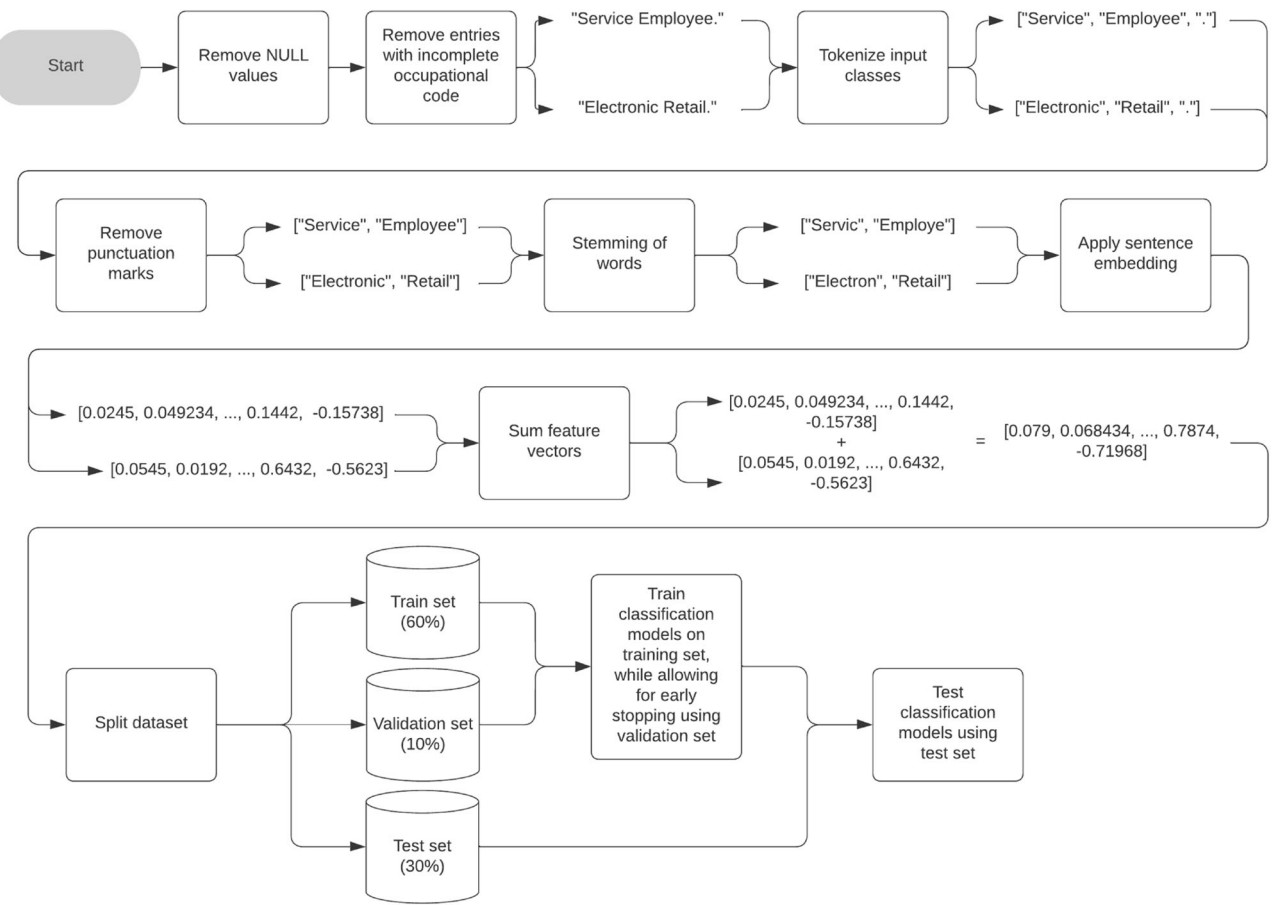

**Fig. 1 Flowchart of the data preparation and training strategy of OPERAS' classification models.** An arbitrary example (i.e., "Service Employee." and "Electronic Retail.'') of two input classes (i.e., job description and sector) of the same entry in an occupational database is given.

classes that yield the best predictive performance for each dataset. Here, we used an exhaustive wrapper approach, where a classification model is trained for each combination of input classes in a dataset and evaluated to find the combination which provides the best predictive performance[38]. Using exclusively the test set, we measured the performance of the classification models.

**Evaluation metrics**. Using the entries in the test set of the corresponding datasets, we assessed the accuracy and inter-coder reliability of OPERAS' classification models using accuracy and Cohen's Kappa ($\kappa$), respectively. Here, we consider the manually expert-coded job descriptions from the aforementioned datasets as the gold standard.

We deem accuracy to be an important metric in the current study, as it indicates the proportion of occupational codes which will potentially require less coding time during the occupational coding process. We define accuracy as the number of correctly predicted codes relative to the total number of predicted codes[39,40].

$$Accuracy\,(\%) = \frac{Number\ of\ correct\ predictions}{Total\ number\ of\ predictions} \times 100\% \quad (1)$$

For each classification model, we computed the accuracy per code level and major occupational group.

Since OPERAS autonomously codes the entries from the test set, it can be considered an (automatic) coder. Hence, OPERAS' inter-coder reliability can be calculated using $\kappa$, which takes into account the correct prediction occurring by chance. This is

defined as:

$$(P_o - P_e)/(1 - P_e), \quad (2)$$

where $P_o$ represents the probability of overall agreement over the label assignments between the classifier and true process. $P_e$ represents the chance agreement over the labels. This is defined as the sum of the proportion of examples assigned to a class times the proportion of true labels of that class in the dataset[41].

OPERAS contains a feature to automatically process predicted codes with a confidence score above a pre-defined threshold. This confidence score is obtained through XGBoost's class probability prediction function[35]. As little to no time will be spent reviewing the codes above the threshold, we grouped predicted codes based on their confidence score in intervals of 5% ranging from 0% to 100% and evaluated the accuracy of each group. Here, we estimate a minimum workload reduction by correcting the automatically processed codes above a threshold for the percentage of correct codes.

**Human-model cross-validation**. We compared OPERAS' inter-coder reliability to two occupational inter-coder reliability studies[9,11]. Maaz et al.[9] studied the inter-coder reliability of two professional coding institutions and two in-house expert-coders in Germany. Here, the self-reported occupations of 300 students' mother and father were coded ($2 \times n = 300$) into ISCO-88 codes, resulting in $n = 12$ coder pairs. They found mean inter-coder reliability (in $\kappa$) on the first, second, third, and fourth coding level of 0.71 (range: 0.68–0.76; SD = 0.04), 0.66 (range: 0.63–0.72; SD = 0.03), 0.57 (range: 0.53–0.63; SD = 0.04), and 0.51 (range: 0.48–0.57; SD = 0.03), respectively. Similar results were found in

Massing et al.[11] where occupational data from a German social survey ($n = 5,130$) and a German field test for adult competencies ($n = 4,159$) (ALLBUS and PIAAC) were separately coded into ISCO-08 by three and two agencies, respectively. This resulted in $n = 4$ coder pairs, where they found mean inter-coder reliability (in $\kappa$) of 0.65 (range: 0.59–0.70; SD = 0.03), 0.60 (range: 0.56–0.65; SD = 0.02), 0.53 (range: 0.46–0.59; SD = 0.03), and 0.46 (range: 0.40–0.55; SD = 0.04), on the same coding levels, respectively.

Using an independent sample $t$-test, we compared the mean inter-coder reliability of each coder pair from these studies to OPERAS' inter-coder reliability on all coding levels. Here, we provide a two-sided $p$ value where the statistical significance level was set at $p < 0.05$. To account for the difference in coding levels and to ensure a fair comparison, we used coding levels 2-5 for the NAF classification model. Given the small sample size, we report the effect sizes in Hedges $g$[42].

**Exposure assessment evaluation**. Using the Formaldehyde-JEM[43] and Silica-JEM[44], we evaluated the exposure assessment accuracy and $\kappa$ for the NAF and PCS classification models. For the ISCO-88, and ISCO-68 classification models, we performed the exposure assessment evaluation using the ALOHA-JEM[45] and DOM-JEM[46], respectively.

We performed the evaluation for two groups, 1) all individuals and 2) exposed individuals. For the first group, we calculated the accuracy using all individuals in the test set. For the second group, we only included job episodes in the evaluation that were deemed exposed according to the gold-standard expert codes. For each JEM, we considered an episode exposed if the total exposure level is higher than 0. This was done to account for the chance level of a non-exposure assignment being substantially higher given the many occupational codes in the JEM resulting in no exposure. Furthermore, assigning an exposed individual a not-exposed status often has the largest effect on the following epidemiological study. Hence, performing an evaluation for this group will give more insight into the applicability of these classification models in real-world scenarios. The accuracy and $\kappa$ in both groups are calculated using Eq. (1) and Eq. (2) respectively. Additionally, we grouped the predicted exposure assessments on the confidence score of the original code in intervals of 5% ranging from 0% to 100% and evaluated the accuracy of each group. To resemble OPERAS' real-world usage of the automatic coding function, we used the minimum confidence score of the NAF or PCS code for the Formaldehyde-JEM and Silica-JEM. In the following sections, we provide additional information on the content and application of the JEMs.

*Formaldehyde-JEM*. The Formaldehyde-JEM has been developed by "Santé Publique France" (the French National Health Surveillance Agency), as part of its "Matgéné" program to assess formaldehyde exposure in the French population[43]. It has been developed by a team of occupational experts through a meta-analysis of 469 scientific, medical, and technical sources. This resulted in a matrix evaluating formaldehyde exposure based on occupational history describing occupation, sector, and corresponding dates between 1950 and 2018. The occupational history was coded using the NAF2008 and PCS2003 and provides three exposure indices for each combination: the probability of exposure, the intensity of exposure, and the frequency of exposure. These respectively refer to the percentage of exposed workers, the mean exposure dose during tasks, and the percentage of working time performing tasks with exposure. Additionally, exposure indices are provided for different calendar periods to account for

variations due to changes in exposure over time. As this study concerned a meta-review, no approval by an Institutional Review Board was needed.

We applied the Formaldehyde-JEM to the gold standard manually coded NAF and PCS codes and OPERAS' predictions to obtain the formaldehyde exposure for each job episode. This is obtained by multiplying the probability, intensity, and frequency of exposure. We used these to calculate the exposure assessment accuracy of OPERAS' NAF and PCS classification models. Further, after applying a JEM to manually coded occupational codes stemming from the same job activity, similar exposure levels are expected[47]. Here, even unequal exposure levels from these codes are still considered relevant for subsequent occupational epidemiological studies. Hence, to assess OPERAS' performance in such real-world exposure assessment scenarios, we also performed the evaluation using a dichotomous status of 'exposed' or 'not exposed'. Further, we measured the rank correlation between the formaldehyde exposure levels of gold-standard manual coding and OPERAS' coding using the Kendall rank correlation coefficient ($\tau$)[48].

*Silica-JEM*. The Silica-JEM assesses the crystalline silica exposure in the French population[44]. Similar to the Formaldehyde-JEM, it has been developed by a team of occupational experts from "Santé Publique France", as part of its "Matgéné" program through a meta-analysis of 469 scientific, medical, and technical sources. This resulted in a matrix evaluating exposure to crystalline silica based on occupational histories describing occupation, sector, and corresponding dates between 1947 and 2007. Similar exposure indices, namely the probability, intensity, and frequency of exposure are provided for each combination of NAF2008 and PCS2003 codes for different calendar periods. As this study concerned a meta-review, no approval by an Institutional Review Board was needed.

For each job episode, we applied the Silica-JEM to the gold standard manually coded and OPERAS-predicted NAF and PCS codes. To obtain the total exposure, we multiplied the probability, intensity, and frequency of exposure and used this to calculate the exposure assessment accuracy. Here, we also calculated the accuracy and $\kappa$ for the dichotomous 'exposed' or 'not exposed' status and calculated $\tau$ for the silica exposure levels.

*ALOHA-JEM*. The ALOHA-JEM has been developed for a study assessing the association of occupational exposure and symptoms of chronic bronchitis and pulmonary ventilatory defects in a general population-based study of five areas in Spain[45]. Subjects completed a respiratory questionnaire on symptoms and occupation and underwent baseline spirometry. Based on expert knowledge, The JEM was developed ad hoc by two occupational experts. It contains a job axis, and an exposure estimate for 12 generic industrial exposures, namely: biological dust, mineral dust, gas fumes, VGDF (i.e., vapors, gases, dust, and fumes), all pesticides, herbicides, insecticides, fungicides, aromatic solvents, chlorinated solvents, other types of solvents, and metals. Exposure estimates for each aforementioned exposure can be obtained for all ISCO-88 codes. Here, three levels of exposure can be assigned, namely 0: no exposure, 1: low exposure, and 2: high exposure. This study protocol was approved by the Institutional Review Board of the participating centers. All patients gave written informed consent.

For each gold standard manually coded and OPERAS-predicted ISCO-88 code, we applied the ALOHA-JEM to obtain a level of exposure for each aforementioned exposure. Here, we deemed OPERAS' exposure assessment correct if all exposures matched this gold standard for each job episode. Consequently, if a single exposure level is inaccurate, it will deem the entire

**Table 4 Per coding level (CL), the outcome categories (OC), accuracy (Acc., %), and inter-coder reliability (Cohen's Kappa, κ) are given for the NAF, PCS, ISCO-88, and ISCO-68 classification models.**

|    | NAF | | | PCS | | | ISCO-88 | | | ISCO-68 | | |
|----|-----|------|------|-----|------|------|---------|------|------|---------|------|------|
| CL | OC | Acc. | κ | OC | Acc. | κ | OC | Acc. | κ | OC | Acc. | κ |
| 1 | 21 | 88.28 | 0.85 | 8 | 84.42 | 0.79 | 10 | 76.50 | 0.73 | 8 | 70.91 | 0.66 |
| 2 | 88 | 84.74 | 0.84 | 24 | 77.23 | 0.76 | 28 | 72.69 | 0.71 | 83 | 63.71 | 0.62 |
| 3 | 272 | 81.80 | 0.81 | 42 | 72.47 | 0.72 | 116 | 66.82 | 0.65 | 284 | 60.88 | 0.60 |
| 4 | 615 | 79.57 | 0.79 | 497 | 68.79 | 0.69 | 390 | 60.95 | 0.60 | 1,506 | 58.31 | 0.57 |
| 5 | 732 | 78.94 | 0.78 | - | - | - | - | - | - | - | - | - |

exposure assessment incorrect. This provides a baseline measure of performance. To allow for the evaluation and comparison of single exposures, we calculated accuracy and κ individually for each exposure. Additionally, we calculated accuracy per exposure level for each exposure in the JEM.

*DOM-JEM.* The DOM-JEM has been developed for the assessment of the inter-method reliability of retrospective exposure assessment related to occupational carcinogens[46]. Based on a multi-center lung cancer case-control study conducted in seven European countries, it contains estimations of nine related exposures for each ISCO-68 code. These are asbestos, chromium, DME, nickel, Polycyclic Aromatic Hydrocarbons (PAH), silica, animals, biological dust, and endotoxin. The DOM-JEM has been developed based on expert knowledge by three independent occupational exposure experts assigning exposure levels to each manually coded ISCO-68 code from the original study. Similar to the ALOHA-JEM, exposure levels ranging 0–2 were used, respectively signifying no exposure, low exposure, and high exposure. This study was approved by the ethics review board of the International Agency for Research on Cancer. Informed consent was obtained for all participants.

We applied the DOM-JEM to the gold standard manually and OPERAS-predicted ISCO-68 codes to obtain a level of exposure for each exposure in the DOM-JEM. Here, we deemed OPERAS' exposure assessment correct if it matched the gold standard for all exposures. Similar to the evaluation conducted with the ALOHA-JEM, we determined overall accuracy and κ for each exposure and computed accuracy per exposure level.

**Reporting summary**. Further information on research design is available in the Nature Portfolio Reporting Summary linked to this article.

## Ethics statement
Since the datasets used in the current study do not contain any information that could be used to identify a person, ethical review from an Institutional Review Board was not required.

## Results
After data cleaning, 281,418 (Constances)[23], 483,090 (Constances), 36,007 (Asialymph)[24], and 12,007 (Lifework)[25] entries were available to train NAF, PCS, ISCO-88, and ISCO-68 classification models, respectively. Additionally, using the Formaldehyde-JEM (NAF&PCS)[43], Silica-JEM (NAF&PCS)[44], ALOHA-JEM (ISCO-88)[45] and DOM-JEM (ISCO-68)[46] exposure assessment evaluation was conducted (see Supplementary Tables S1 and S2). This evaluation was conducted for two groups: all individuals, providing OPERAS' performance in real-world exposure assessment scenarios, and exposed individuals, to account for the chance level of a non-exposure assignment being substantially higher.

Using the expert codes as the gold standard, we used prediction accuracy and Cohen's Kappa (κ) as evaluation metrics (see Table 4 and Supplementary Tables S3-S6). Additionally, OPERAS provides a confidence score indicating the probability of correctness of a suggested code for each prediction. This is used to interpret the quality and usefulness of a classification model (see Fig. 2).

**Model evaluation**. *NAF and PCS* use Constances' input-classes occupation and sector with an accuracy per major occupational group ranging between 52.17–95.91% and 32.45–90.56%, respectively. 57.41% of the suggested NAF codes have a confidence score in the 95–100% range, with an accuracy of 97.02%. 43.11% of the suggested PCS codes are within this range, with 94.47% correctly predicted. Using the Formaldehyde-JEM and Silica-JEM, exposure assessment for all individuals gave 98.09% (κ = 0.84) and 98.41% (κ = 0.67) accuracy, respectively. Here, 31.28% of the suggested codes had a 95–100% minimum confidence score for both the Formaldehyde-JEM and Silica-JEM, with 99.32% and 99.77% accurate exposure assessment, respectively. Evaluation for the dichotomous 'exposed' or 'not exposed' status for this group gave 98.30% (κ = 0.85) and 98.51% (κ = 0.68) accurate exposure assessment for the Formaldehyde-JEM and Silica-JEM, respectively. Kendall rank correlation for the exposure levels of the same JEMs respectively was τ = 0.85 (p < 0.01) and 0.68 (p < 0.01). For exposed individuals (6.30% and 2.58% of job episodes, respectively), exposure assessment using the same JEMs respectively gave 81.85% (κ = 0.81) and 61.82% (κ = 0.61) accuracy. 54.22% and 20.27% of these codes had a 95–100% confidence score, of which 98.59% and 93.33% had correct exposure assessment, respectively. Here, the dichotomous evaluation resulted in 85.18% and 65.54% accurate exposure assessment for the Formaldehyde-JEM and Silica-JEM, respectively.

*ISCO-88* uses Asialymph's input-classes occupation, task, employer, and product with an accuracy per major occupational group ranging between 63.53–96.11%. 21.54% of the predicted codes have a confidence score in the 95–100% range, with 94.57% accuracy. Using the ALOHA-JEM, exposure assessment for all individuals gave 75.05% accuracy (range: 83.69–98.03%) and κ = 0.70, where 21.54% of the suggested codes had a 95–100% confidence score with 97.68% correct exposure assessment. For the exposed individuals (62.53% of job episodes), exposure assessment gave 65.58% (range: 57.12–87.68%) accuracy, κ = 0.63, with 22.35% of the suggested codes having a 95–100% confidence score, where 96.88% had a correct exposure assessment.

*ISCO-68* uses Lifework's job name input-class, where the accuracy per major occupational group ranged 53.58–80.67%. 20.65% of the suggested codes have a 95–100% confidence score, with a 95.38% correct prediction. Using the DOM-JEM, exposure assessment for all participants gave 84.19% accuracy (range: 91.84–98.90%) and κ = 0.59 where 20.65% of the suggested codes

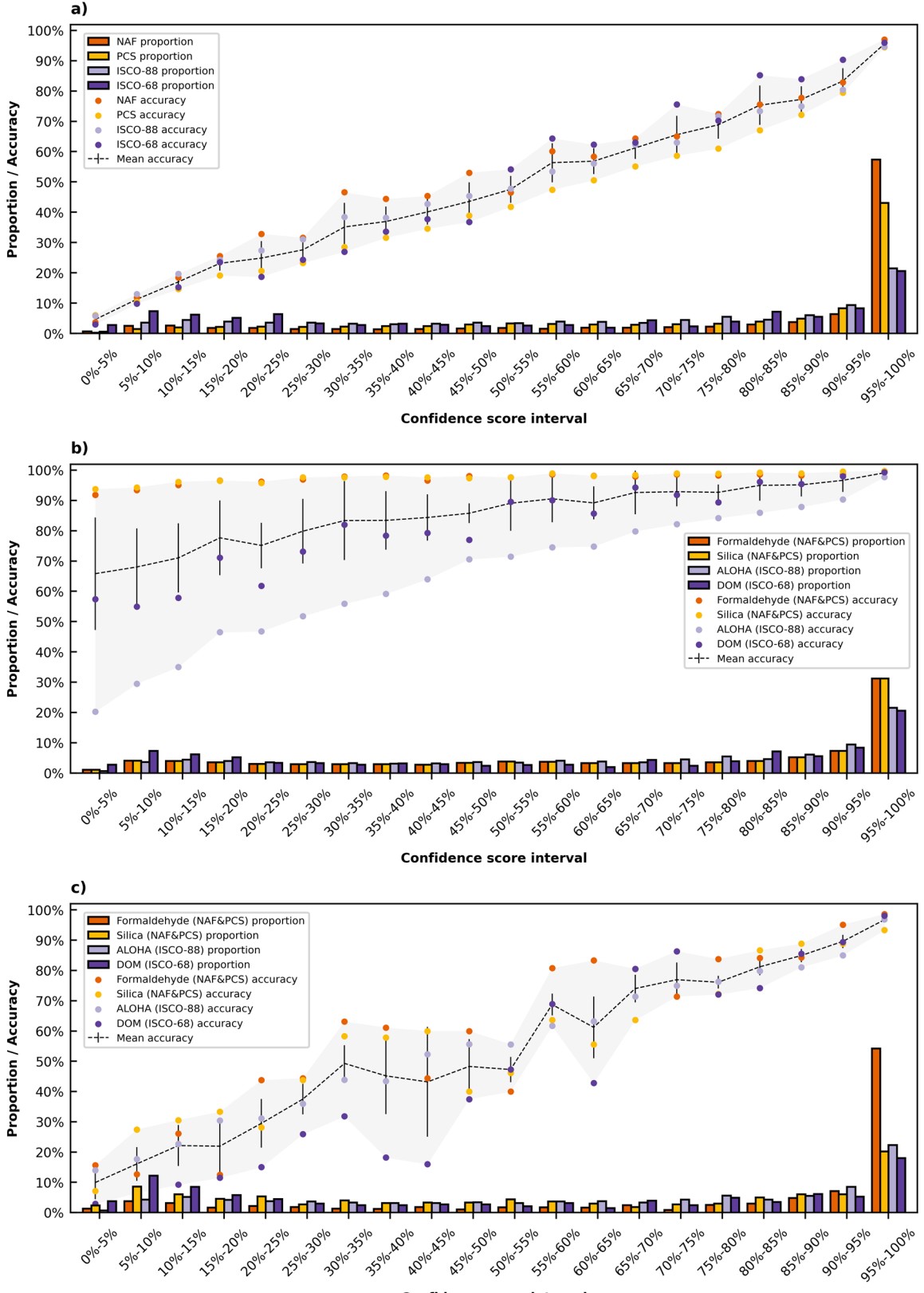

**Fig. 2 Confidence score distribution and accuracy of the classification performance and exposure assessment evaluation.** The proportion (%) of codes in a confidence score interval is denoted by the bars in the histogram. The mean accuracy (%) and standard deviation of codes ($n = 4$ independent accuracy scores) in an interval are respectively represented by the dotted and vertical lines in the continuous graph. Here, the dots denote the individual accuracy of each classification model in an interval. The grey area represents the accuracy range over the intervals. Colors correspond to the different classification models. **a** Results of the classification performance evaluation. **b** Results of the exposure assessment evaluation of all individuals and **c** exposed individuals.

**Table 5 Per coding level (CL⁺) the mean, range, standard deviation (SD), and comparison (including Hedges g) of expert coders (N = 16) with OPERAS (N = 4) are given.**

| CL[a] | Expert coders | | | OPERAS | | | t-test | | |
|---|---|---|---|---|---|---|---|---|---|
| | Mean | Range | SD | Mean | Range | SD | t(df) | p | g |
| 1 | 0.67 | 0.59–0.76 | 0.04 | 0.75 | 0.66–0.84 | 0.08 | 3.17 (18.00) | 0.005 | 0.36 |
| 2 | 0.62 | 0.56–0.71 | 0.04 | 0.72 | 0.62–0.81 | 0.08 | 3.68 (18.00) | 0.002 | 0.43 |
| 3 | 0.54 | 0.46–0.63 | 0.04 | 0.69 | 0.60–0.79 | 0.08 | 3.53 (3.37) | 0.032 | 0.62 |
| 4 | 0.47 | 0.40–0.56 | 0.05 | 0.65 | 0.57–0.78 | 0.08 | 3.71 (3.36) | 0.028 | 0.64 |

[a]For NAF, CL 2-5 were used (see Table 4).

had a 95–100% confidence score, with 99.32% correct exposure assessment. For the exposed individuals (24.54% of job episodes), exposure assessment gave 49.97% (range: 25.00–59.71%) accuracy and $\kappa = 0.46$, with 17.98% of the suggested codes having a 95–100% confidence score and 98.11% correct exposure assessment.

**Human-model cross-validation.** Human inter-coder reliability studies including coder pair level inter-coder reliabilities report ranges (in $\kappa$) of 0.51–0.71[9] and 0.46–0.65[11]. We compared this to OPERAS' inter-coder reliability (see Table 4).

The Shapiro-Wilk Test for Normality and Levene's Test for Equality of Variances reveal that all coding levels are normally distributed, where coding levels 3 and 4 do not have equal variances across groups. Hence, to compare OPERAS' inter-coder reliability to the aforementioned human inter-coder reliability studies we use the independent sample $t$-test while correcting for the inequality of variances of coding levels 3 and 4. We find that OPERAS significantly outperforms expert coders on all coding levels (see Table 5).

**Human workload reduction.** OPERAS improves the efficiency of the occupational coding process by enabling the automated processing of codes with a confidence score above a custom threshold. Figure 2 shows the confidence score distribution of the NAF, PCS, ISCO-88, and ISCO-68 classification models in intervals of 5%, ranging from 0% to 100%. The (mean) prediction accuracy within each interval is also displayed.

An estimated minimum workload reduction can be calculated through the percentage of automatically coded job descriptions above a threshold while correcting for incorrectly coded job descriptions. Given a confidence score threshold of 95% for the NAF, which has 57.4% of the suggested codes above this threshold of which 97.0% are correct, a minimum workload reduction of 55.7% ($57.41 \times 0.9702$) could be realized. Although the overall prediction accuracy of the PCS, ISCO-88, and ISCO-68 classification models is lower, their accuracy within the 95–100% confidence score interval remains similar. Hence, given the same confidence score threshold, a minimum workload reduction of 40.7%, 20.4%, and 19.7% could be realized using OPERAS' PCS, ISCO-88, and ISCO-68 classification models, respectively.

## Discussion

To build robust generic classification models, we refrained from tweaking hyperparameter settings. Instead, we used default values and included generalized job descriptions from the coding indexes in the training data[22]. This provided us with a good baseline performance that can be expected to generalize to other datasets[36]. Further, OPERAS proposes a limited set of probable codes for similar job descriptions. This holds the potential for improved inter-coder reliability among expert coders utilizing OPERAS[13], which can contribute to the occupational coding process' robustness. Moreover, OPERAS' adaptable automatic coding can enable a substantial workload reduction for a wide range of applications. For example, the default 95% confidence score threshold could be lowered to increase workload reduction for applications that are tolerant to less accurate codes or solely require high accuracy on lower coding levels.

Differences in classification performance can be attributed to the difference in available job descriptions per classification model. The NAF and PCS were respectively trained with 281,418 and 483,090 entries, whereas the ISCO-88 and ISCO-68 models were trained with respectively 36,007 and 12,007 entries. Additionally, the difference in combined job description length between datasets could also have contributed to the difference in classification performance. Longer job descriptions decrease coding reliability if the additional information does not directly correspond to the definition of the occupational code[12]. This is due to an increase in potential occupational codes that can be assigned to a job description. In classification model training, the information increases variance within outcome categories, resulting in decreased classification performance[49]. The accuracy of OPERAS' classification models shows a comparable non-monotonic downward trend with an increase in description length. However, the very small number of entries for the larger description lengths of the used input classes (see Table 3) warrants being careful with definitive conclusions on the association between prediction accuracy and description length.

The difference in exposure assessment performance could be attributed to the Formaldehyde-JEM and Silica-JEM containing specific occupational exposures that are relatively rare in the general population. Hence, the chance of discordant codes resulting in a non-exposed status is relatively high[50]. When the prevalence of exposed individuals increases, or when only exposed individuals are considered in the comparison, inter-rater reliability tends to decrease (see Fig. 2b, c). However, a moderate increase in reliability can be observed during the dichotomous evaluation for exposed individuals of the Formaldehyde-JEM and Silica-JEM. The correlation coefficients for all exposure levels also show strong and moderate correlation for respectively the same JEMs. However, they are not near perfect as evidenced by their inter-rater reliability for all individuals. Hence, to ensure reliable health outcomes it is crucial to take these factors into account during (semi-automatic) occupational coding[47,50].

The class imbalance in current datasets presents a substantial challenge for the development and deployment of the classification models. This is due to the fact that ML models tend to over-classify the majority class, thereby compromising the classification performance of minority classes[32]. This is a large issue, particularly in the current domain where the misclassification of minority classes can have critical implications on health outcomes. Traditional methods designed to tackle this issue, such as SMOTE, are in this case inadequate as the representation of the minority class in the datasets is excessively low[33,51]. As such, it is

important that alternative data augmentation techniques are developed that can effectively enhance the representation of minority classes in datasets where the representation of the minority class is (close to) one. This will contribute to a more balanced and accurate performance of ML models, thereby increasing their generalizability and applicability in critical domains.

OPERAS' classification models are trained using one occupational dataset each. However, job description characteristics often differ between occupational datasets, possibly resulting in a lower out-of-distribution performance[11,20]. Additionally, OPERAS has been trained and tested on expert-coded occupational data coded on the least aggregated level. Although OPERAS' data pre-processing pipeline can manage and produce predictions for ambiguous or incomplete data, its performance in such scenarios remains untested. Hence, to assess the generalizability of OPERAS' classification models, its evaluation could be extended using external validation sets. To further improve the classification model performances, other classification techniques, hyper-parameter settings, and the use of additional datasets could be considered[52]. Additionally, OPERAS' human-model cross-validation is currently based on external occupational inter-coder reliability studies containing different occupational characteristics compared to the current ones[9,11,23–25]. Hence, to gain more insight into OPERAS' performance against expert-coders, additional cross-validations using the same underlying datasets could be performed.

With OPERAS, a decision support system for epidemiological job coding of free-text job descriptions is introduced, which outperforms both expert coders[9,11] and state-of-the-art coding tools[18,53]. This is achieved through the ML-based classification models for four (inter)national classification systems. OPERAS' codes are accompanied by confidence scores that can be used by the expert coder to partially automate the coding process with accurate exposure assessment, resulting in a substantial workload reduction. Additional insight into OPERAS' classification performance can be gained through the use of external data sets and comparisons with expert coders using the same underlying data. As such, OPERAS supports custom occupational coding, enabling large-scale occupational health research in an efficient, effective, accurate, and stable manner.

## Data availability

The Constances[23], Asialymph[24], and Lifework[25] datasets are not publicly available. Access to these datasets should be requested from the authors of the original studies. The Formaldehyde[43] and Silica[44] JEMs can be consulted on the Exp-pro portal of "Santé publique France" at https://exppro.santepubliquefrance.fr/matgene. Access to the computer files of the matrices for their use in the context of epidemiological studies should be requested from the authors of the original study. The ALOHA[45] and DOM[46] JEMs are not publicly available and should be requested from the authors of the original study. Source data underlying the graphs in Fig. 2 are available as Supplementary Data 1.

## Code availability

The development and evaluation code of the current classification models is available at https://zenodo.org/records/8390811[54]. The used test sets from the Constances, Asialymph, and Lifework datasets can only be obtained with permission from the authors of the original studies. The code has been developed in Python 3.8 using the following Python packages: flair 0.8.0.post1, nltk 3.6.2, numpy 1.19.5, pandas 1.2.4, scikit-learn 0.24.1, tokenizers 0.10.2, and XGBoost 1.5.2. The OPERAS software is still in development and future versions might use classification models utilizing different packages and techniques than the ones currently described.

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

## Acknowledgements

The authors thank the Constances team for providing the job histories of the cohort participants and Fabien Gilbert (IRSET, University of Angers) for supervising their coding. The authors would also like to thank the National Cancer Institute for providing the AsiaLymph job descriptions. The OPERAS project discloses support for the research of this work from ANSES [PNR EST-2018/1/106]. The Constances cohort is supported and funded by the Caisse Nationale d'Assurance Maladie (CNAM). It benefits from a grant from the Agence nationale de la recherche [ANR-11-INBS-0002] and from the French Ministry of Research. Lifework was funded by the Netherlands Organisation for Health Research and Development (ZonMw) within the Electromagnetic Fields and Health Research programme [85200001] and [85500003]. S.P. is supported by the Exposome Project for Health and Occupational Research (EPHOR) which is funded by the European Union's Horizon 2020 research and innovation programme [874703]. R.C.H.V. is supported by the Gravitation program of the Dutch Ministry of Education, Culture, and Science and the Netherlands Organization for Scientific Research through the EXPOSOME-NL [024.004.017]. N.R., Q.L., and M.F. were supported by the Intramural Research Program of the Division of Cancer Epidemiology and Genetics, NCI, NIH.

## Author contributions

Conceptualization: M.A.L., E.L.v.d.B., S.P., M.G., G.R., R.C.H.V. Data curation: M.A.L. Formal Analysis: M.A.L. Funding acquisition: E.L.v.d.B., S.P., M.G., R.C.H.V. Investigation: M.A.L. Methodology: M.A.L., E.L.v.d.B. Project administration: E.L.v.d.B., S.P., M.G., R.C.H.V. Resources: S.P., M.G., S.L., N.R., Q.L., R.C.H.V. Software: M.A.L. Supervision: E.L.v.d.B., S.P., M.G., G.R., R.C.H.V. Validation: M.A.L. Visualization: M.A.L. Writing—original draft: M.A.L., E.L.v.d.B. Writing—review and editing: M.A.L., E.L.v.d.B., S.P., M.G., G.R., M.F., R.C.H.V.

## Competing interests

The authors declare no competing interests.
