## [Peer Review File · Communications Medicine]

Reviewers' comments:

Reviewer #1 (Remarks to the Author):

Generic comments

The manuscript describes an effort aimed at predicting standard occupational job codes using machine learning techniques based on several large databases. The paper describes the model used and its performance in terms of finding the right job code and assigning the right exposure status. While I believe this is a very important and interesting work, overall I found the manuscript would need some rebalancing with currently too much description of the inner workings of the model, which was in fact described elsewhere, and not enough on the calculation of the performance metrics, in particular those related to exposure assessment.

It is a very personal opinion, but I also found the tone a tad too boastful for a scientific manuscript (e.g. the short title “ AI surpasses...” shown on every page, the abstract reporting only the highest ratings, the end of the introduction already stating the study was a great success...).

Section 2.3: much too much space for an algorithm described elsewhere

Section 2.3 takes much too much space in my opinion. The authors clearly state they have used a model already described elsewhere (ref 13) but they nevertheless take almost 5 pages to describe the intricacies of the model. Sometimes it is useful to describe the principles of an approach known in other fields as an introduction to a new field, but this description seem to me much too technical for that type of aim. I strongly suggest spending much less time on section 2.3 (from 4+ to less than one) and better document other aspects of the methods (the exposure assessment for example, or the natural language pre-treatment : “sentence embedding” this could be explained better for this reader).

Section 2.5 : OPERAS intercoder study

I can't find any information on the opera inter-coder studies? (maybe I missed them) : sample size, selection of specific (e.g. exposed) occupations or random sample, training of the coders....L544 suggest the population is the test set but it would be nice to have the info more clearly available.

And is it a subsample of jobs or a subsample of individuals and their job history?

Moreover, I think the ANOVA analysis really brings very little to the table, the raw numbers seem amply sufficient to me.

Finally, the SDs for the OPERAS column on table 5 seem off (we would expect the range close to $\text{mean} + 2 \cdot \text{SD}$, which works for the expert column but not for the OPERAS column). If this is not a mistake, this is further argument against the ANOVA.

Section 2.6 : exposure assessment.

This section could be presented much more clearly. I would suggest one section per JEM describing the metrics in details in addition to what is already available in the manuscript. E.g. : DOM JEM provide a semi-quantitative per occupational code from 0, not exposed, to 3 (which is a strike to the mention of probability of exposure in JEMs in the intro).

For example, it was very unclear what the metrics for the formaldehyde and silica JEMs were: how did the authors calculate a cumulative exposure index over one job ? (Or did they calculate exposure for subjects as opposed to individual jobs ?, and if they used individuals, how did they average an unexposed job with a job with a rating of 2 ?). Please describe the raw JEM metrics and the final exposure metrics from which kappa or other agreement metrics were calculated. To use a sentence on the journal's guidance to reviewers, this validation effort could not be reproduced by independent researchers at it stands. Also, provide the proportion of unexposed (range for example) as this is a major determinant of agreement.

Discussion

The discussion feels a bit perfunctory and lacking a clear structure (e.g. modelling, coding reliability, exposure assessment accuracy, productivity gain, limitations)

For example, the single sentence about difference in exposure assessment performance is insufficient in my opinion to the point of being almost meaningless given the little details provided in other sections of the manuscript. The “large proportion of unexposed” issue (clearly apparent in figure 2) is important and should be better documented/discussed. It is amply discussed for example in Rémen T, Richardson L, Siemiatycki J, Lavoué J. (2021) Impact of Variability in Job Coding on Reliability in Exposure Estimates Obtained via a Job-Exposure Matrix. Ann Work Expo Health. PMID: 34931220.

The section on job description length could also be improved and I believe it would be possible for the authors to provide empirical data from their own datasets about the association of description length with performance (a bit like they stratified performance by prediction certainty category they could stratify by description length).

I am also a bit curious about the assertion that using the default parameters in the Machine learning models will improve “out of distribution” performance. I have very little knowledge in this field but usually technical people tend to criticize unfamiliar users using default parameters.

Specific comments

Summary: the result section of the summary feels very small. Moreover, the authors chose to report the maximal values (“up to”, “as far as”). I think it would be more appropriate to report ranges and the median, maybe IQRs too.

L093: clarify the applicability of the number, does it include “prolonged sitting”, is it in the whole word, on the Netherland?

L93-95: check syntax

L097: clarity, “in population based cohorts”

L104-106: this would go better in the methods. Also provide an example of hierarchy, e.g. manufacturing/footwear/sport shoe manufacture or something like that

L122-L134: several remarks on that section. “k score” not defined. The sentence about lower reliability but higher exposure assessment accuracy of humans is not supported by the numbers shown in my opinion, especially since they seem to come from different studies: human: 36-50% rel + k between 0.66-0.84, computer : 50%, k 0.4-0.8. I don’t believe these numbers, as shown, support the wording of L122-123 or L127-129. In my opinion there are many good reasons to create a tool like OPERAS but the proposed ones on these sentences are weak/doubtful.

L133-135: maybe wait for the discussion/conclusion ?

L220: 1-21.52 and not 1.21.52

L258: low or high variability ?

L258-261: I could not understand what was done

L268-275: please discuss this limitation in the discussion part.

L481: presumably, for occupational coding, for which the number of outcomes is very large (hundreds of categories), P_e would consistently be almost zero, what’s the added value compared to accuracy ? The various tables clearly show the redundancy of the 2 metrics (for occupational coding). Maybe choose only one?

L498 : mention type of population, country and sample size please.

L692: please clarify this somewhat strange assertion.

L694-698: this is an interesting result. I suggest its determination deserves a small methods section and a bit longer results section. I would also discuss it a bit in the discussion section.

Table1: add country consistently to the classification name, add references to the results

Table 2: suggest using median and IQR instead of metrics assuming a normal distribution, an

obviously wrong assumption

Table 3: suggest eliminating mean and SD (see previous comment) and add Tukeys' five numbers (quantiles 0,0.25,0.5,0.75,1)

Figure 2 : very clever and efficient presentation

References : the following work might be judged relevant by the authors (or not), the first on occupational coding reliability, the second on the effect of that reliability on exposure assessment through JEMs

Rémen T, Richardson L, Siemiatycki J, Lavoué J. (2021) Impact of Variability in Job Coding on Reliability in Exposure Estimates Obtained via a Job-Exposure Matrix. *Ann Work Expo Health*. 2021 Dec 21;wxab106. doi: 10.1093/annweh/wxab106. Epub ahead of print. PMID: 34931220

Rémen, T., Richardson, L., Pilorget, C., Palmer, G., Siemiatycki, J., Lavoué, J. (2018) Development of a coding and crosswalk tool for occupations and industries. *Annals of work exposures and health* 62(7):796-807. PMID:29912270

Review performed by Jérôme Lavoué, Professor, University of Montréal.

Reviewer #2 (Remarks to the Author):

I think that this manuscript can be published as it is now. The researchers have done a great, sound work.

Reviewer #3 (Remarks to the Author):

This is an important paper which is worthy of publication. However, I do have some reservations about the way in which the information is processed and the results presented, notably in terms of the treatment of free text descriptions, and the way in which job codes have already been assigned to the data sets concerned.

Datasets from three countries form the basic information that is used for the study of the classification algorithm termed OPERAS. The paper describes how variables from these datasets are pre-processed prior to coding. This involves a very significant reduction for the French dataset ('Constances') with no information included in the text as to why this is the case, apart from the statement that they were insufficiently detailed descriptions. If coding tools are to prove useful, they must be tested using the types of information that are generated in statistical enquiries.

The paper references inter-coder reliability at various points, noting that the use of coding algorithms improves reliability between different coders. This is undoubtedly true, given that most software coding tools suggest a small number of potential codes and (usually) ranks these via a probabilistic scoring mechanism. Given that different coders using such tools are then faced with the same choice sets, this will improve reliability between coders. The critical question to be answered though is 'do such software coding tools improve accuracy?'

Accuracy is measured via the use of so-called 'gold standard' coded datasets. These are data that have generally been coded once by manual or semi-automated techniques, then checked very carefully by expert coders, who may have access to further information that was not made available

to the original coders. Thus, for example, a code may be assigned to a vague job title such as 'engineer', but detailed inspection of sector and qualifications may result in this being recoded to a professional engineering occupation or a skilled manual occupation. This involves the development and use of what are termed default codes – codes which are to be used in situations where the quality of the text to be coded does not result in a clear allocation to one specific category. Although the paper refers to the data that are used as being coded via a process 'performed manually by expert coders', this does not make clear to the reader that this is what is recognised as 'gold standard'. In the absence of this information, it is difficult to accept the term 'accuracy' as it is currently used in the paper. More information is required about how the three datasets were initially coded.

The discussion section of the paper raises an interesting point, relating to the amount of information that is presented to the coding algorithm. For the PCS information in Constances, this is clearly just a job title. We know little about how this text was originally coded, but it is most likely that some form of default coding was used. Also, the point is well made that the amount of information that is presented to the algorithm increases, the coding reliability decreases. This is a well-known observation, that has resulted in the adoption of coding methods where just job titles are processed first, then those that have failed to reach some threshold score are coded in a semi-automated fashion, taking account of any additional information that may improve coding accuracy.

I have some minor points as follows:

The title – this uses the term 'artificial intelligence', yet the intelligence that is used is very human and comes from the way in which the algorithm attempts to mimic human decision making. We use similar models for such activities as weather forecasting but would not consider that a weather forecast is somehow the product of an artificial intelligence. Can the authors come up with a more apposite title?

Abstract, 3rd line – what does 'harmonized' mean?

Abstract, 4th line – 'fully' is redundant.

Methods, 4th line – spell out the names of the classifications.

Plan language summary, 1st line - replace 'can' with 'may'.

Plan language summary, 10th line – delete the hyperbole 'state-of-the-art' (also from the 4th paragraph).

Introduction, 2nd paragraph, 9th line – not all the classifications used are hierarchically structured.

Data preparation, 2nd paragraph, 7th line – replace 'retainal' with 'retention'.

Discussion, 2nd paragraph, 3rd line – '482,090' versus '483,090' in Table 2.

Discussion, 2nd paragraph, 13th line – delete 'superfluous'.

Dear referees,

Thank you for your careful consideration of the manuscript. We greatly appreciate your thoughtful and constructive feedback. We are confident that this has improved the quality of the manuscript. We have processed all comments. In this letter, we address them point-by-point, noting the locations of relevant changes at the end of each response. Additionally, while carefully processing your feedback, we have made some additional minor textual improvements. Please see the revised manuscript for details. We hope all changes are in line with your expectations.

Thank you again for your consideration of the manuscript.

On behalf of our co-authors, yours faithfully,

Egon L. van den Broek and Mathijs A. Langezaal

Reviewer one

(1) General comment on the manuscript:

It is a very personal opinion, but I also found the tone a tad too boastful for a scientific manuscript (e.g. the short title “ AI surpasses...” shown on every page, the abstract reporting only the highest ratings, the end of the introduction already stating the study was a great success...).

Response:

- We understand the comment. We have discussed it among the authors, dropped one of the claims in the title, and focussed on the main claim. Additionally, we opted for a less hyperbolic choice of words. We hope this is in line with the thought of the reviewer. If not, we will reverse this change. *(L006)*
 - The title is shown on each page due to the used LaTeX template. In the final print, the title will only be shown on the first page of the manuscript, solving this issue.
 - The format of Communications Medicine requires a short summary of the results at the end of the introduction section. However, indeed the wording can be perceived as boastful. Hence, we rephrased the final sentence of the introduction. *(L169-172)*
 - We now report numeric results (inter-coder reliability ranges, exposure assessment accuracy ranges, etc.) in the abstract. *(L059-064)*
-

(2) Comment:

Section 2.3: much too much space for an algorithm described elsewhere

Section 2.3 takes much too much space in my opinion. The authors clearly state they have used a model already described elsewhere (ref 13) but they nevertheless take almost 5 pages to describe the intricacies of the model. Sometimes it is useful to describe the principles of an approach known in other fields as an introduction to a new field, but this description seem to me much too technical for that type of aim. I strongly suggest spending much less time on section 2.3 (from 4+ to less than one) and better document other aspects of the methods (the exposure assessment for example, or the natural language pre-treatment : “sentence embedding” this could be explained better for this reader).

Response:

- We agree with these comments. We have adapted a large part of section 2.3 to reduce its size and technicality, where only parts of the algorithm applied to the current application have been reported. To ensure a complete overview of the algorithm for interested readers, we moved the extended description of the algorithm to the supplementary information. *(L316-355, Supplementary Information)*
 - Indeed, sentence embeddings are an important aspect of the algorithm, especially since these allow similar job descriptions to be described as such. Hence, we have provided more explanation in section 2.2. *(L285-292)*
-

(3) Comment:

Section 2.5 : OPERAS intercoder study

I can't find any information on the opera inter-coder studies? (maybe I missed them) : sample size, selection of specific (e.g. exposed) occupations or random sample, training of the coders....

L544 suggest the population is the test set but it would be nice to have the info more clearly available. And is it a subsample of jobs or a subsample of individuals and their job history?

Moreover, I think the ANOVA analysis really brings very little to the table, the raw numbers seem amply sufficient to me.

Finally, the SDs for the OPERAS column on table 5 seem off (we would expect the range close to mean+2*SD, which works for the expert column but not for the OPERAS column). If this is not a mistake, this is further argument against the ANOVA.

Response:

- In case the reviewer is referring to "OPERAS' inter-coder reliability' in L498 of the original manuscript, these are Cohen's Kappa values of OPERAS' classification models. Since the codes are autonomously generated/coded, OPERAS could be considered an (automatic) coder. Consequently, Since the test set contains 'gold standard' expert-coded job descriptions, the inter-coder reliability can be calculated. In case the reviewer is referring to 'two occupational inter-coder reliability studies' (which we find unlikely), these are external inter-coder reliability studies (as seen by ref 9 and 11). We have provided additional clarification in section 2.4. (L384, 397-400)
- We have performed the Shapiro-Wilk Test for Normality and Levene's Test for Equality of Variances and found that all data is normally distributed. However, coding levels 3 and 4 do not have equal variances across groups. Hence, we have opted to instead use the independent sample t-test as a statistical test, while correcting for this inequality of variance of levels 3 and 4. (L436, 441-442, 677-684, Table 5)

(4) Comment:

Section 2.6 : exposure assessment.

This section could be presented much more clearly. I would suggest one section per JEM describing the metrics in details in addition to what is already available in the manuscript. E.g. : DOM JEM provide a semi-quantitative per occupational code from 0, not exposed, to 3 (which is a strike to the mention of probability of exposure in JEMs in the intro).

For example, it was very unclear what the metrics for the formaldehyde and silica JEMs were: how did the authors calculate a cumulative exposure index over one job ? (Or did they calculate exposure for subjects as opposed to individual jobs ?, and if they used individuals, how did they average an unexposed job with a job with a rating of 2 ?). Please describe the raw JEM metrics and the final exposure metrics from which kappa or other agreement metrics were calculated. To use a sentence on the journal's guidance to reviewers, this validation effort could not be reproduced by independent researchers at it stands. Also, provide the proportion of unexposed (range for example) as this is a major determinant of agreement.

Response:

- The DOM and ALOHA-JEM indeed do not provide a probability of exposure. However, in the manuscript, this statement is only related to the Formaldehyde and Silica JEMs. This indeed suggests that the structure of this section is unclear and hence, we followed your

recommendation to write a section for each JEM where additional information will be given. (L467-541)

- Regarding the Formaldehyde and Silica we calculated the exposure for individual job episodes. The agreement metrics are calculated on the basis of whether the exposure level is the same for the predicted and gold-standard codes. As previously mentioned, we further explained this in the separate JEM sections. (L484-490, 499-503, 521-524, 538-541)
 - In addition to the number of exposed individuals provided in Supplementary Table S1, we provide the proportion of exposed job episodes in the results section. (L580, 591, 668)
 - We have fixed an error in the discussion section. Previously, we had mistakenly stated that the DOM-JEM contained exposures that are seldom found in the population. The correct statement now refers to the Formaldehyde-JEM instead. (L741)
-

(5) Comment:

The discussion feels a bit perfunctory and lacking a clear structure (e.g. modelling, coding reliability, exposure assessment accuracy, productivity gain, limitations)

For example, the single sentence about difference in exposure assessment performance is insufficient in my opinion to the point of being almost meaningless given the little details provided in other sections of the manuscript. The “large proportion of unexposed” issue (clearly apparent in figure 2) is important and should be better documented/discussed. It is amply discussed for example in Rémen T, Richardson L, Siemiatycki J, Lavoué J. (2021) Impact of Variability in Job Coding on Reliability in Exposure Estimates Obtained via a Job-Exposure Matrix. Ann Work Expo Health. PMID: 34931220.

Response:

- We improved the structure, where clear distinctions between the different subjects should now be more apparent. (L705-791)
 - We now provide additional discussion material on the exposure assessment including the provided additional reference. (L742-748)
-

(6) Comment:

The section on job description length could also be improved and I believe it would be possible for the authors to provide empirical data from their own datasets about the association of description length with performance (a bit like they stratified performance by prediction certainty category they could stratify by description length).

Response:

- Indeed, this will further strengthen this point. Hence, we performed an additional evaluation using OPERAS’ datasets and classification models. (L731-739)
-

(7) Comment:

I am also a bit curious about the assertion that using the default parameters in the Machine learning models will improve “out of distribution” performance. I have very little knowledge in this field but usually technical people tend to criticize unfamiliar users using default parameters.

Response:

- In most cases, a model is finetuned based on a specific dataset. This process optimizes the model's capacity to process inputs and generate outputs specifically for that particular dataset. If out-of-distribution (OoD) datasets are highly similar to the original, this could be seen as a benefit. However, when it comes to occupational datasets, they often exhibit diverse job description representations. As a result, if a model is excessively finetuned to one dataset, its performance may likely diminish when dealing with OoD data. To substantiate this, we provide an additional reference. (L361, 711, ref [36])

(8) Comment:

Summary: the result section of the summary feels very small. Moreover, the authors chose to report the maximal values (“up to”, “as far as”). I think it would be more appropriate to report ranges and the median, maybe IQRs too.

Response:

- This is due to the limited word count. However, to extend the results sections without exceeding the word limit, we have opted to reduce the number of words in the background and methods section. (L042-043, removed first statement in Methods, L052-054)
- Regarding the ranges, we agree. Hence, we now report the ranges. (L059-064)

(9) Comment:

L093: clarify the applicability of the number, does it include “prolonged sitting”, is it in the whole word, on the Netherland?

Response:

- We have rewritten and extended the sentence to provide additional clarification. As the reference was indicative of the Netherlands, we have replaced it with a reference on the global effect of occupation on health. (L094-097, ref [2])

(10) Comment:

L93-95: check syntax

Response:

- Indeed, the sentence is a bit odd. We have rewritten this. (L097-100)
-

(11) Comment:

L097: clarity, "in population based cohorts"

Response:

- We have added "In population-based cohorts" to the sentence. (L101)
-

(12) Comment:

L104-106: this would go better in the methods. Also provide an example of hierarchy, e.g. manufacturing/footwear/sport shoe manufacture or something like that

Response:

- We agree with this statement. Hence, we removed the explanation about the hierarchy of occupational codes from the introduction section. Additionally, we have added an example of the hierarchy in the first paragraph of section 2.1. (L210-212)
-

(13) Comment:

L122-L134: several remarks on that section. "k score" not defined. The sentence about lower reliability but higher exposure assessment accuracy of humans is not supported by the numbers shown in my opinion, especially since they seem to come from different studies:

- human: 36-50% rel + k between 0.66-0.84,
- computer : 50%, k 0.4-0.8.

I don't believe these numbers, as shown, support the wording of L122-123 or L127-129. In my opinion there are many good reasons to create a tool like OPERAS but the proposed ones on these sentences are weak/doubtful.

Response:

- k score refers to Cohen's Kappa. We clarified this in the manuscript. (L128, 130)
 - We agree that the highest Cohen's Kappa score for automatic coding tools is comparable to that of human coders. Nevertheless, it is noteworthy that the lowest Cohen's Kappa score for human coders surpasses that of automatic tools by a margin of 0.26. This serves as an indication that human coders, even while possessing equivalent or marginally lesser inter-coder reliability, are capable of ensuring a more consistent exposure assessment than automatic occupational coding tools. This suggests that for reliable exposure assessment results, human participation in the coding process remains vital. Hence, we have opted to keep this reason and add small clarifications in the manuscript. (L129, 131)
 - In line with the reviewer's comment, we have added additional reasons for the development of OPERAS. (L133-167, 780-784)
-

(14) Comment:

L133-135: maybe wait for the discussion/conclusion ?

Response:

- This is due to the required format. However, we adapted the wording of the sentence as per comment (1). *(L169-172)*
-

(15) Comment:

L220: 1-21.52 and not 1.21.52

Response:

- Indeed this is incorrect. We have fixed this. *(L260)*
 - Further, we noticed while processing this comment that the provided ISCO-68 code is non-existent. Hence, we changed it to "1-21.10". *(L260)*
-

(16) Comment:

L258: low or high variability ?

Response:

- High is meant here. It has been added for clarification. *(L281)*
-

(17) Comment:

L258-261: I could not understand what was done

Response:

- We provided additional clarification and details on the usage of the sentence embeddings. *(L285-292)*
-

(18) Comment:

L268-275: please discuss this limitation in the discussion part.

Response:

- Indeed, the fact that data augmentation such as SMOTE cannot be used is an important limitation. We have added a paragraph on this limitation in the discussion section. *(L749-761)*
-

(19) Comment:

L481: presumably, for occupational coding, for which the number of outcomes is very large (hundreds of categories), Pe would consistently be almost zero, what's the added value compared to accuracy? The various tables clearly show the redundancy of the 2 metrics (for occupational coding). Maybe choose only one?

Response:

- Indeed, for the least aggregated occupational levels of the classifications Pe will be almost zero. However, this is not the case for higher coding levels with fewer outcome categories. The addition of Cohen's kappa also provides an additional metric for comparison with other studies. Without this metric, human-model cross-validation is also not possible. Hence, we believe Cohen's kappa remains a valuable metric to report.

(20) Comment:

L498 : mention type of population, country and sample size please.

Response:

- This is indeed relevant information. We have added this to the first paragraph of section 2.5. (L421-424, 428-432)

(21) Comment:

L692: please clarify this somewhat strange assertion.

Response:

- We rewrote the final sentence of this section to be clearer and more indicative of the aforementioned results. (L682-683)

(22) Comment:

L694-698: this is an interesting result. I suggest its determination deserves a small methods section and a bit longer results section. I would also discuss it a bit in the discussion section.

Response:

- We have added additional clarification on the confidence score and estimated workload reduction calculation in the final paragraph of section 2.4. Furthermore, we have added section 3.3 on workload reduction and discussed this in the discussion section. (L414-416, L685, 688-703, 714-719)
-

(23) Comment:

Table1: add country consistently to the classification name, add references to the results

Response:

- We have added both to Table 1. (*Table 1, L139-162*)
-

(24) Comment:

Table 2: suggest using median and IQR instead of metrics assuming a normal distribution, an obviously wrong assumption

Response:

- We agree that the mean and SD are indeed not the best metrics to report. Hence, we now report the median and IQR. (*Table 2, L231-239*)
-

(25) Comment:

Table 3: suggest eliminating mean and SD (see previous comment) and add Tukeys' five numbers (quantiles 0,0.25,0.5,0.75,1)

Response:

- Similar to the previous response, we used this instead. (*Table 3, L242-248*)
-

(26) Comment:

References : the following work might be judged relevant by the authors (or not), the first on occupational coding reliability, the second on the effect of that reliability on exposure assessment through JEMs

Rémen T, Richardson L, Siemiatycki J, Lavoué J. (2021) Impact of Variability in Job Coding on Reliability in Exposure Estimates Obtained via a Job-Exposure Matrix. *Ann Work Expo Health*. 2021 Dec 21:wxab106. doi: 10.1093/annweh/wxab106. Epub ahead of print. PMID: 34931220

Rémen, T., Richardson, L., Pilorget, C., Palmer, G., Siemiatycki, J., Lavoué, J. (2018) Development of a coding and crosswalk tool for occupations and industries. *Annals of work exposures and health* 62(7):796-807. PMID:29912270

Response:

- Indeed, both are relevant references. Thank you for bringing this to our attention. We added the first reference in the discussion section on the effect of job description length on exposure assessment accuracy. (*L742-743, ref. [48]*)
 - We added the second reference in the introduction section paragraph where coding tools are discussed. Initially, our focus was solely on automatic coding tools. However, recognizing the continued importance of manual coding tools like CAPS, we felt it necessary to broaden our scope. Therefore, we have added additional references regarding tools that facilitate manual coding. (*L121-123, ref. [13, 14]*)
-

(27) Comment:

Figure 2 : very clever and efficient presentation

Response:

- We sincerely appreciate the reviewer's positive feedback on the figure! Hard work went into its creation and presentation, so the compliment is truly valued!
-

Reviewer 2

(1) General comment on the manuscript:

I think that this manuscript can be published as it is now. The researchers have done a great, sound work.

Response:

- We thank the reviewer for this great compliment on our work! We have put a lot of effort into the study and its manuscript, and therefore greatly appreciate the comment!
-

(27) Comment:

Datasets from three countries form the basic information that is used for the study of the classification algorithm termed OPERAS. The paper describes how variables from these datasets are pre-processed prior to coding. This involves a very significant reduction for the French dataset ('Constances') with no information included in the text as to why this is the case, apart from the statement that they were insufficiently detailed descriptions. If coding tools are to prove useful, they must be tested using the types of information that are generated in statistical enquiries.

Response:

- Indeed, that is the case. However, currently, we want OPERAS to suggest full codes to the user. Hence, job descriptions that were not coded to the highest coding level were removed for training. However, vague or incomplete job descriptions can still be coded using OPERAS. This however will likely result in suggested codes with a lower confidence score. Since we do agree with the statement, we added additional information on this in the discussion section. (L765-768)
-

(28) Comment:

The paper references inter-coder reliability at various points, noting that the use of coding algorithms improves reliability between different coders. This is undoubtedly true, given that most software coding tools suggest a small number of potential codes and (usually) ranks these via a probabilistic scoring mechanism. Given that different coders using such tools are then faced with the same choice sets, this will improve reliability between coders. The critical question to be answered though is 'do such software coding tools improve accuracy?'

Response:

- We completely agree with this statement! In occupational coding, 'accuracy' is a hard-to-define term, since no gold standard exists. The closest we get to a 'gold standard' for each occupational classification are the examples listed in the coding index for every occupational code. Consequently, job descriptions that can't be directly coded to these examples do not have a definitive code. This is where expert coders come into play, as they could be considered the 'gold standard' in this scenario.

Since OPERAS can autonomously code job descriptions when provided with the input, you could refer to OPERAS as an (automatic) coder. Hence, when we measure the output of OPERAS against the 'gold standard' expert-coded job descriptions in the test set, we can calculate the performance metrics. This is what we refer to in the manuscript as inter-coder reliability and accuracy. However, we, unfortunately, did not measure if OPERAS improves inter-coder reliability between two coders, as this would require two (or more) expert coders to code a dataset of job descriptions using OPERAS. Since this is a very interesting and important consequence of these types of automatic coding tools, we added information on this in the discussion section. (L711-714)

(29) Comment:

Accuracy is measured via the use of so-called 'gold standard' coded datasets. These are data that have generally been coded once by manual or semi-automated techniques, then checked very carefully by expert coders, who may have access to further information that was not made available to the original coders. Thus, for example, a code may be assigned to a vague job title such as 'engineer', but detailed inspection of sector and qualifications may result in this being recoded to a professional engineering occupation or a skilled manual occupation. This involves the development and use of what are termed default codes – codes which are to be used in situations where the quality of the text to be coded does not result in a clear allocation to one specific category. Although the paper refers to the data that are used as being coded via a process 'performed manually by expert coders', this does not make clear to the reader that this is what is recognised as 'gold standard'. In the absence of this information, it is difficult to accept the term 'accuracy' as it is currently used in the paper. More information is required about how the three datasets were initially coded.

Response:

- Regarding the accuracy and gold standard, we agree, which we also mentioned in our response to comment (28). Hence, we clarified throughout the manuscript what we considered and used as gold standard when accuracy was computed. (L058-059, 384-386, 484, 499, 520, 524, 538, 541)
- We provided additional information on the coding of the datasets. (L217-224, 253-254, 263, 264)

(30) Comment:

The discussion section of the paper raises an interesting point, relating to the amount of information that is presented to the coding algorithm. For the PCS information in Constances, this is clearly just a job title. We know little about how this text was originally coded, but it is most likely that some form of default coding was used. Also, the point is well made that the amount of information that is presented to the algorithm increases, the coding reliability decreases. This is a well-known observation, that has resulted in the adoption of coding methods where just job titles are processed first, then those that have failed to reach some threshold score are coded in a semi-automated fashion, taking account of any additional information that may improve coding accuracy.

Response:

- The NAF, PCS, ISCO-88, and ISCO-68 also use the sector or activity as part of the input for the coding algorithm, which has proven to enhance classification performance. However, the current algorithm is flexible enough to accept only the job description if other data is not available. This is possible through sentence embeddings, which ensure similar descriptions are interpreted in a similar way by the algorithm. Still, providing additional information improves the algorithm's performance by increasing the variance between different occupational codes. Additionally, the inclusion of a confidence score allows for semi-automatic coding as described by the reviewer. In the manuscript, we provided further details on the effect of job description length on classification performance and sentence embeddings. (L285-292, 731-739)
-

(31) Comment:

The title – this uses the term ‘artificial intelligence’, yet the intelligence that is used is very human and comes from the way in which the algorithm attempts to mimic human decision making. We use similar models for such activities as weather forecasting but would not consider that a weather forecast is somehow the product of an artificial intelligence. Can the authors come up with a more apposite title?

Response:

- The current methods are inherently machine learning techniques (i.e., gradient tree boosting), which is a subset of artificial intelligence. Hence, we felt it was important to reflect the AI aspects of our methodologies in the title and thus opt to use the term artificial intelligence in the title. However, we do acknowledge that the classification of certain algorithms as artificial intelligence can be a subject of debate in certain instances.

(32) Comment:

Abstract, 3rd line – what does ‘harmonized’ mean?

Response:

- In this sentence, harmonized means the same as coded. Hence, it is redundant and has been removed. (L044)

(33) Comment:

Abstract, 4th line – ‘fully’ is redundant.

Response:

- Indeed, we removed this. (L045)

(34) Comment:

Methods, 4th line – spell out the names of the classifications.

Response:

- Indeed, the classifications should be spelled out. Hence, we did this. (L050-052)

(35) Comment:

Plan language summary, 1st line - replace ‘can’ with ‘may’.

Response:

- This indeed improves the readability of the sentence. Hence, we replaced can with may. (L071)
-

(36) Comment:

Plan language summary, 10th line – delete the hyperbole ‘state-of-the-art’ (also from the 4th paragraph).

Response:

- We have removed hyperboles such as ‘state-of-the-art’ regarding the current work with ranges or less hyperbolic language throughout the manuscript. (L079-080, 169-172)
-

(37) Comment:

Introduction, 2nd paragraph, 9th line – not all the classifications used are hierarchically structured.

Response:

- To our knowledge, all mentioned classifications are hierarchically structured. However, the next number/letter is not always the next level in the hierarchy. We clarified this in the manuscript. Hence, we have removed this assertion from the introduction section and added additional clarification in section 2.1. (L209)
-

(38) Comment:

Data preparation, 2nd paragraph, 7th line – replace ‘retainal’ with ‘retention’.

Response:

- Indeed, this is incorrect. We changed this to improve readability. (L282)
-

(39) Comment:

Discussion, 2nd paragraph, 3rd line – ‘482,090’ versus ‘483,090’ in Table 2.

Response:

- This is indeed a typo. We used 483,090, since that is the correct number. (L722)
-

(40) Comment:

Discussion, 2nd paragraph, 13th line – delete ‘superfluous’.

Response:

- We have removed ‘superfluous’ from this sentence. (L729-730)
-

Reviewers' comments:

Reviewer #1 (Remarks to the Author):

Generic comments

The authors have mostly answered my comments.

I still have one significant gripe with the exposure assessment evaluation, which came out as the authors gave more details about this aspect:

First with the definition of the “exposed” group (for the analysis restricted to exposed jobs) : exposed according to whom ? What if the occupational code suggested by OPERAS said “exposed” and the code chosen by the human code (the gold standard if I understood correctly) said “unexposed” ? how was the exposure status selected ? Please clarify (as an illustration, a possible definition would be : one job is defined as exposed only if both OPERAS and the human code result in an exposed code).

Second how was the “exposed” status created from each JEM ? For ALOHA and DOM JEM I guess this is easy : exposed is anything not in the category 0. But what about the formaldehyde and silica JEM, which have probability / intensity / frequency ? is 5 min per week exposed at 1/100 of the OEL in an occupation with a probability of exposure of 5% considered as “exposed” ? please clarify

Third, if I understand correctly what was done, for the quantitative JEMs, the authors calculated for each JEM cell the multiplication : probability*intensity*frequency as an exposure index, then, for calculation of accuracy / kappa of “exposure levels” they created a dichotomous variable “agree” if the OPERAS exposure index was equal to the human exposure index. As an exposure assessment person this is very unintuitive to me. Presumably (I don't know the JEMs) these exposure indices can vary across occupations by 2-4 orders of magnitude (probability from 0 to 100%, frequency from 0 to 100%, exposure level maybe from 1/100*OEL to >OEL). Using the authors' approach, the situation of 2 exposure indices 100 for OPERAS and 110 for human coder would lead to the same “DISAGREE” decision as an exposure index of 1 for OPERAS and 500 for human coder. This hardly makes sense to me. What we hope when looking at exposure levels after applying JEMs, is that despite different codes, exposure levels/probability of exposure will be in the same ballpark because the expert coded similar activities. Here, by using this strict equality as a criterion, in a way the authors are just repeating the analysis of the agreement in assigned code: it is indeed quite plausible that there will be close to as many different combinations of probability/exposure intensity/exposure frequency values as there are occupational categories. If I did not grossly misunderstand what was done, this is a very significant limitation of this assessment, which then adds little to the direct coding accuracy assessment.

The issue above might seem less problematic for the ordinal JEMs such as ALOHA and DOM JEM. However, in their case, agreement is defined as the same rating “for all chemicals” : this again will render the comparison close to the comparison of the codes, as there are also probably many combinations of ratings across all chemicals.

I understand the authors had to make decisions as to how to measure agreement in exposure across many datasets and exposure information sources but their approach of “disagreement = exposure

levels not strictly equal” feels off. To my knowledge, other studies have reported on agreement on the exposed/not exposed dichotomous status with kappa, and then, separately for actual exposure levels, correlation metrics such as Kendal Tau or Spearman rho. What the authors call accuracy in exposure assessment isn’t really comparable to other works in the field in my opinion.

Apologies if I misconstrued the actual approach but at least I guess it warrants some clarification.

Minor comments / suggestions:

Correct equation numbers

L185 : add the corresponding sub-code in brackets for each name.

Figure 1 : The 2 expressions that are in the end merged into a single set of numbers, are they two descriptions coming for the same job code, two sentences from the same job description ? (L231 to 251 still hard to understand for me). Additionally, I don’t think Fig 1 is introduced anywhere in the text.

Reviewer #3 (Remarks to the Author):

This is a resubmission, so I confine my comments to state whether the paper is now suitable for publication as amended by the authors.

It is my view that this paper is now ready for publication. There are some minor corrections i would like to see added to the abstract and the discussion . These are listed below:

line 044 – change ‘prevent’ to ‘explore’

line 064 – change ‘accurate’ to ‘a high degree of accuracy’

line 067 – change ‘state-of-the-art’ to ‘other current coding tools’

line 080 – delete ‘automatic’

line 783 - refers to 'four (international) classifications', yet the only two which are international are ISCO88 and ISCO68.

Department of Information and Computing Sciences
Utrecht University
Princetonplein 5
3584 CC Utrecht
the Netherlands

August 18, 2023

Dear referees,

Thank you again for your consideration of the manuscript. We are pleased to hear that most of the changes from the previous revision were found satisfactory. We appreciate the further critical and insightful comments. We were able to process all of them and provide a point-by-point response in this letter. The location of the changes in the manuscript can be found at the end of each response. We hope these are in line with your expectations.

Thank you for your continued consideration of the manuscript.

On behalf of our co-authors, yours faithfully,

 Egon L. van den Broek and Mathijs A. Langezaal

Reviewer 1

(1) Comment:

The authors have mostly answered my comments.

Response:

We are pleased to hear that we have addressed the majority of the reviewer's comments.

(2) Comment:

First with the definition of the “exposed” group (for the analysis restricted to exposed jobs): exposed according to whom? What if the occupational code suggested by OPERAS said “exposed” and the code chosen by the human code (the gold standard if I understood correctly) said “unexposed”? how was the exposure status selected? Please clarify (as an illustration, a possible definition would be: one job is defined as exposed only if both OPERAS and the human code result in an exposed code).

Response:

We indeed regard manual coding as the gold standard. Consequently, we consider a job exposed if the occupational code from the manual coder results in exposure according to the JEM. Subsequently, we calculated the accuracy for this group by comparing it to the exposure assessment following OPERAS’ automatic coding for the same entries. We have now provided additional clarification on this in the methodology section. (L459-464)

(3) Comment:

Second how was the “exposed” status created from each JEM? For ALOHA and DOM JEM I guess this is easy: exposed is anything not in the category 0. But what about the formaldehyde and silica JEM, which have probability / intensity / frequency? is 5 min per week exposed at 1/100 of the OEL in an occupation with a probability of exposure of 5% considered as “exposed”? Please clarify

Response:

Indeed, for the ALOHA&DOM JEMs, an episode is considered exposed if it is not in category 0. For the Formaldehyde and Silica JEM, we consider an episode exposed if the total exposure level is higher than 0. We adopted this definition because this boundary may vary for each subsequent occupational epidemiological study where these JEMs are utilized, and we cannot presume the specific epidemiological studies where OPERAS will be applied in the future. Consequently, we now explicitly mention this boundary of ‘exposed’ or ‘not exposed’ in the methodology section. (L461-462)

(4) Comment:

Third, if I understand correctly what was done, for the quantitative JEMs, the authors calculated for each JEM cell the multiplication: probability*intensity*frequency as an exposure index, then, for calculation of accuracy/kappa of “exposure levels” they created a dichotomous variable “agree” if the OPERAS exposure index was equal to the human exposure index. As an exposure assessment person this is very unintuitive to me. Presumably (I don’t know the JEMs) these exposure indices can

vary across occupations by 2-4 orders of magnitude (probability from 0 to 100%, frequency from 0 to 100%, exposure level maybe from $1/100 \cdot \text{OEL}$ to $>\text{OEL}$). Using the authors' approach, the situation of 2 exposure indices 100 for OPERAS and 110 for human coder would lead to the same "DISAGREE" decision as an exposure index of 1 for OPERAS and 500 for human coder. This hardly makes sense to me. What we hope when looking at exposure levels after applying JEMs, is that despite different codes, exposure levels/probability of exposure will be in the same ballpark because the expert coded similar activities. Here, by using this strict equality as a criterion, in a way the authors are just repeating the analysis of the agreement in assigned code: it is indeed quite plausible that there will be close to as many different combinations of probability/exposure intensity/exposure frequency values as there are occupational categories. If I did not grossly misunderstand what was done, this is a very significant limitation of this assessment, which then adds little to the direct coding accuracy assessment.

The issue above might seem less problematic for the ordinal JEMs such as ALOHA and DOM JEM. However, in their case, agreement is defined as the same rating "for all chemicals": this again will render the comparison close to the comparison of the codes, as there are also probably many combinations of ratings across all chemicals.

I understand the authors had to make decisions as to how to measure agreement in exposure across many datasets and exposure information sources but their approach of "disagreement = exposure levels not strictly equal" feels off. To my knowledge, other studies have reported on agreement on the exposed/not exposed dichotomous status with kappa, and then, separately for actual exposure levels, correlation metrics such as Kendal Tau or Spearman rho. What the authors call accuracy in exposure assessment isn't really comparable to other works in the field in my opinion

Response:

The reviewer's understanding of the methodology is correct. Indeed, we considered OPERAS' exposure estimate to be correct solely if it exactly matches the exposure assessment from the manual coding. This was done to provide a raw/clean performance metric for OPERAS' coding and, subsequently, the exposure assessment. Further, what is considered a 'correct' or 'acceptable' deviation between two coders differs per occupational epidemiological study. With the current methodology, we aimed to provide results that show a 'minimum' performance level; they are an underestimation of the real-world exposure assessment performance, as described by the reviewer.

We do agree with the reviewer that the current methodology indeed hampers the comparability with other works in the field. Hence, we followed the reviewer's recommendation to calculate correlation metrics using Kendall's Tau and perform an evaluation on the dichotomous status 'exposed' or 'not exposed' for the Formaldehyde and Silica JEMs. Further, to make it comparable with studies that applied the ALOHA and DOM JEMs, we now also provide accuracy for each of their individual exposure levels. (L495-505, 518-520, 538-543, 561-563, 578, 657-662, 666-668, 775-780, and Supplementary Table S2)

(5) Comment:

Correct equation numbers

Response:

Thank you for noting this. We have corrected the equation numbers. (L469)

(6) Comment:

L185: add the corresponding sub-code in brackets for each name.

Response:

In the revised manuscript L185 refers to the following sentence: *“Furthermore, to ensure optimal classification performance, we reduced the dimensionality and retained crucial information of the free-text job descriptions through multiple Natural Language Processing (NLP) techniques.”*. However, we are unsure what name and corresponding sub-code the reviewer is referring to. Consequently, we assume the reviewer is referring to L210-214, where, following previous comments, an example of the major, sub-major, minor, and unit groups are now provided for ISCO-88 code “2221”. We believe that the reviewer refers to the name of these groups, where the sub-code for each group should be added. We agree with this and added the sub-codes to each group name. (L213-214)

(7) Comment:

Figure 1: The 2 expressions that are in the end merged into a single set of numbers, are they two descriptions coming for the same job code, two sentences from the same job description? (L231 to 251 still hard to understand for me)

Response:

The two expressions are feature vectors of two sentences (i.e., input classes) of the same job code (i.e., entry). To further clarify this in the manuscript, we have extended the caption of Figure 1 and adapted the explanation of the summing of feature vectors in the methodology section. (L295-297) (L344-346)

(8) Comment:

Additionally, I don't think Fig 1 is introduced anywhere in the text

Response:

Figure 1 is currently introduced in the first sentence of section 2.3 (L313-315). We opted to introduce it here since the Figure encapsulates the entire development of OPERAS' classification models, which is introduced in this sentence.

Reviewer 3

(9) Comment:

It is my view that this paper is now ready for publication.

Response:

We are happy that the reviewer is satisfied with the previous revision and views the paper as publishable.

(10) Comment:

line 044 – change ‘prevent’ to ‘explore’

Response:

Indeed, this is more accurate. Hence, we have changed ‘prevent’ to ‘explore’. *(L044)*

(11) Comment:

line 064 – change ‘accurate’ to ‘a high degree of accuracy’

Response:

We changed ‘accurate’ to ‘a high degree of accuracy in...’. *(L064, 066)*

(12) Comment:

line 067 – change ‘state-of-the-art’ to ‘other current coding tools’.

Response:

We have changed ‘state-of-the-art’ to ‘other current coding tools’. *(L067)*

(13) Comment:

line 080 – delete ‘automatic’

Response:

We deleted ‘automatic’. *(L081)*

(14) Comment:

line 783 - refers to 'four (international) classifications', yet the only two which are international are ISCO88 and ISCO68.

Response:

Indeed, this is the case. Thank you for noting this. We changed “(international)” to “(inter)national”, to represent both the national (i.e., PCS and NAF) and international (i.e., ISCO-88 and ISCO-68)

classification systems. This is also more in line with our previous description of the classification systems in the introduction section. (L815)

REVIEWERS' COMMENTS:

Reviewer #1 (Remarks to the Author):

The authors have adequately answered previous comments